# Knowledge, Attitudes, and Practices Regarding Air Pollution among Medical Students

**DOI:** 10.3390/ijerph21060789

**Published:** 2024-06-17

**Authors:** Santiago Rendon-Marin, Luis Felipe Higuita-Gutiérrez, Diana Maryory Gomez-Gallego

**Affiliations:** 1Infettare, Facultad de Medicina, Universidad Cooperativa de Colombia, Medellín 050012, Colombia; santiago.rendon@campusucc.edu.co; 2Facultad de Medicina, Universidad Cooperativa de Colombia, Medellín 050012, Colombia; luis.higuita@udea.edu.co; 3Escuela de Microbiología, Universidad de Antioquia, Medellín 050010, Colombia

**Keywords:** air pollution, medical students, attitudes, practices, knowledge

## Abstract

*Background:* Air pollution has emerged as a global public health concern. Specifically, in Medellín, Colombia, episodes of elevated air pollution have been documented. Medical students’ knowledge of air pollution is paramount for implementing future interventions directed toward patients. The aim of this research was to delineate the knowledge, attitudes, and practices regarding air pollution among medical students at a private university in Medellín. *Methods:* A cross-sectional study involving 352 medical students was conducted. A questionnaire was administered, generating scores ranging from 0 to 100, where a higher score signified better knowledge, attitudes, and practices. Data were analyzed using frequencies, summary measures, non-parametric tests, and linear regression. *Results*: In total, 31% rated the education received at the university on the relationship between health and air quality as fair to poor, and 81% perceived the air quality in the city as poor. The knowledge score was 77.8 (IQR 71.1–85.6), with 90% acknowledging that exposure to air pollution increases the risk of various diseases. The attitudes score was 82.1 (IQR 71.8–87.2), and 25.9% believed that air pollution is a multifactorial problem, rendering their actions ineffective. In terms of practices, the score was 50 (IQR 42.9–57.1), indicating that students either did not employ protective measures against pollution or used inappropriate practices such as masks or air purifiers. Regression analysis revealed no association between knowledge and practices. *Conclusion:* The findings of this study underscore that medical students possess commendable knowledge regarding the health effects of air pollution. However, their adoption of inappropriate practices for self-protection is evident. The lack of correlation between knowledge and practices highlights the necessity of educational initiatives to be complemented by regulatory and cultural interventions.

## 1. Introduction

Clean air is regarded as a fundamental principle for maintaining health. However, due to rapid population and industrial growth in major urban centers, air pollution levels have surged. The World Health Organization (WHO) reported that 99% of the global population resides in regions where exposure to air pollution surpasses recommended limits. Furthermore, the WHO warned that by the year 2019, approximately 4.2 million premature deaths worldwide would be attributable to air pollution, with nearly 90% of these deaths occurring in low- and middle-income countries [1,2]. This underscores a disproportionate disease burden due to air pollution in populations facing heightened vulnerability [3]. Air pollution is assessed by measuring the quantity of particles (particulate matter (PM)) and gaseous components in the air. PM is defined as solid and liquid substances in the atmosphere, while gaseous pollutants include carbon monoxide, volatile organic compounds, nitrogen oxides, and ozone [4,5,6]. Outdoor air pollution originates from both natural and anthropogenic sources. Natural sources include volcanic eruptions, wildfires, dust storms, and biological emissions from plants and soil. On the other hand, anthropogenic sources are human activities that contribute significantly to outdoor air pollution. These sources include industrial emissions, vehicle exhaust, power generation from fossil fuels, agriculture (e.g., livestock operations and fertilizer use), and residential combustion of solid fuels such as wood and coal [7,8]; however, it is also imperative to consider indoor air pollution originating primarily from combustion processes (e.g., cooking and heating) using solid fuels, wood burning, tobacco smoke, building materials, organic solvents in household and personal care products, and other domestic consumer goods [3,9,10,11].

Air pollution can lead to various adverse health effects [6,12]. It has been documented to increase the risk of cardiovascular diseases, such as heart conditions, hypertension, atherosclerosis, acute myocardial infarction, and cerebrovascular accidents [13,14]. Additionally, it is associated with respiratory diseases, including lung cancer, chronic obstructive pulmonary disease (COPD), and asthma [8,14]. Moreover, air pollution has been linked to neurological disorders, such as cognitive function loss, anxiety, autism, and diseases like Parkinson’s and Alzheimer’s [15,16]. Inflammatory skin conditions, such as atopic dermatitis, acne, psoriasis, and allergic reactions, are also among the health issues attributed to air pollution [17]. Furthermore, it may cause complications in fetal development, including premature birth and low birth weight [18].

Air pollution is one of the most concerning environmental issues for Colombians due to the impacts on health and the environment. Additionally, it is the third highest factor contributing to social costs after water pollution and natural disasters. In Colombia, according to the latest study by the National Institute of Health (INS, Instituto Nacional de Salud), 15,681 deaths are associated with poor air quality [19]. Furthermore, the National Planning Department (DNP, Departamento Nacional de Planeación) estimated that the annual costs associated with deaths caused by poor air quality are over COP 12 billion. This is because deaths occur due to chronic obstructive pulmonary disease, acute respiratory infections, and lung cancer, which are expensive to treat [20].

In recent years, the city of Medellín, Colombia, has witnessed recurring episodes of air pollution which are adversely affecting the health of its residents and leading to the city being ranked as one of the most polluted in Latin America [21,22]. According to a study conducted between 2010 and 2016 in Medellín, years of healthy life adjusted for disability (DALYs) related to exposure to environmental PM constituted 13.8% of the city’s total burden, with 71.4% of these DALYs attributed to mortality, particularly among individuals aged over 65 years [21]. Contributing factors to air pollution in the city include topography, meteorological conditions, thermal inversion events, population growth, and an increase in the vehicular fleet [23,24]. All these factors contribute to atmospheric contingency episodes in Medellín, particularly between February and April. During these emergency periods, restrictive measures are implemented for all vehicular categories, fixed industrial sources emitting more than 100 milligrams per cubic meter of PM are prohibited from operating, and the general community is advised to refrain from outdoor activities and to engage in telecommuting [25,26].

Given the magnitude and consequences of this issue, various interventions have been designed to reduce air pollution and its health effects [27]. Among all available interventions, efforts have been made to increase awareness within the community regarding the high morbidity and mortality burdens attributable to air pollution and its primary emission sources. Promoting literacy among the population regarding the impact of air pollution on health, climate, and the environment is a crucial strategy to address the problem [3]. In this context, healthcare professionals play a fundamental role. On the one hand, they can advocate for public policies to prevent or mitigate the risks of air pollution to human health. On a clinical level, they can inform and raise awareness among patients about these risks and appropriate strategies for protection [28,29]. Therefore, physicians and medical students’ knowledge, attitudes, and practices regarding air pollution are fundamental for implementing counseling programs for the general population on this topic.

Studies on “knowledge, attitudes, and practices” (KAP) aim to elucidate what is known, believed, and done concerning a particular topic. Data from KAP-type surveys are essential in the planning, implementation, and evaluation of a program or strategy, as well as in identifying knowledge gaps, cultural beliefs, or behavioral patterns that may facilitate or hinder the success of that strategy [30]. Although multiple publications of this nature, conducted with physicians and healthcare professionals, assessing their preparedness, knowledge, perceptions, and practices regarding pollution, climate change in general, and their impacts on health [28,29,31,32], exist globally, there is a notable absence of similar research conducted on medical students in the city of Medellín, Colombia, despite the significance that this issue holds for future physicians in this city. Thus, the present study aimed to describe the knowledge, attitudes, and practices regarding air pollution among medical students in Medellín. Specifically, the study aimed to answer the research question: How do medical students in Medellín, Colombia, perceive and respond to issues related to air pollution, including their knowledge levels, attitudes, and actual practices concerning air quality protection and mitigation strategies? The results of this study could provide valuable information for designing strategies in the curriculum of medical programs and offering enhanced education on environmental pollution to students.

## 2. Materials and Methods

### 2.1. Hypothesis

Based on the knowledge, attitudes, and practices (KAP) framework [30], our study hypothesizes that medical students’ knowledge about air pollution sources, health effects, and mitigation strategies (knowledge) will positively correlate with their attitudes toward air pollution and its impact on health and the environment (attitudes), which in turn will influence their actual practices related to air pollution prevention and protection (practices).

### 2.2. Study Type

This research employed a descriptive cross-sectional design.

### 2.3. Study Population

The study population comprised all medical students enrolled in a private university in Medellín, Colombia. The sample size was calculated based on a reference population of 1100 students, a confidence level of 95%, a design effect of 1, a precision of 1, and an expected standard deviation on the KAP scale of 12 points. According to these parameters, it was determined that 369 students needed to be included. Participants of both genders, above legal age, and from all academic semesters who willingly agreed to participate were included in the study. In the sampling process, an effort was made to ensure representation from students across all semesters as follows: first-semester *n* = 30 (8.5%), second-semester *n* = 40 (11.4%), third-semester *n* = 42 (11.9%), fourth-semester *n* = 39 (11.1%), fifth-semester *n* = 34 (9.7%), sixth-semester *n* = 6 (1.7%), seventh-semester *n* = 8 (2.3%), eighth-semester *n* = 21 (6%), ninth-semester *n* = 25 (7.1%), tenth-semester *n* = 69 (19.6%), eleventh-semester *n* = 19 (5.4%), twelfth-semester *n* = 19 (5.4%). In total, 352 students were successfully included.

### 2.4. Data Collection Instrument

Based on information gathered from other KAP studies on air pollution [29,32,33,34], an initial questionnaire was designed to assess the variables of interest. This questionnaire consisted of 71 questions, which underwent refinement through expert reviews and pilot testing, leading to the development of the final instrument. The final data collection form was divided into four parts: (1) sociodemographic variables and perceptions of air pollution, comprising 9 items; (2) a questionnaire on knowledge about air pollution with 15 items, including questions about air pollution sources, their health effects, and mitigation strategies; (3) a questionnaire on attitudes toward air pollution with 13 items; (4) a questionnaire on practices related to air pollution with 14 items (see the Appendix A for the survey). A Likert scale with four levels, ranging from completely disagree to completely agree, was employed for the knowledge and attitudes indices. The practices index Likert scale included four response options from never to always. The instrument was designed to present each item individually or as part of an aggregated scale for the knowledge, attitudes, and practices indices. A score was generated for each index ranging from 0 (minimum possible score) to 100 (maximum possible score) using the following formula:
$$\begin{array}{c} \mathrm{Index}\mathrm{Score}=\\ \left[\left(\mathrm{Score}\mathrm{obtained}-\mathrm{Minimum}\mathrm{possible}\mathrm{score}\right)/\left(\mathrm{Maximum}\mathrm{possible}\mathrm{score}-\mathrm{Minimum}\mathrm{possible}\mathrm{score}\right)\right]\times 100 \end{array}$$

### 2.5. Data Collection

Data collection employed three strategies: (i) institutional emails, (ii) classroom visits, and (iii) invitations to course instructors to encourage questionnaire responses. In brief, institutional emails were used to invite medical students from a private university in the city to participate in the study. Invitations were extended to all students to ensure representation from all semesters until the predetermined sample size was achieved. Simultaneously, requests were made for opportunities to address some classes personally, inviting students to participate and complete the questionnaire. The self-administered online form included informed consent and could only be filled out by those students who willingly agreed to participate in the research.

### 2.6. Data Analysis

Demographic characteristics of the students, perceptions of the education received, and the perception of air quality in the city were described using absolute and relative frequencies. Age is presented as the mean and standard deviation. In contrast, the score for each index is presented with the median and interquartile range due to the lack of compliance with normality assumption. Comparisons between index scores based on demographic characteristics, perception of education received, and perception of air quality were conducted using the Mann–Whitney U and Kruskal–Wallis H tests. Finally, linear regressions were performed to identify factors associated with knowledge, attitudes, and practices and potential confounding variables. Regression models were assessed for assumptions of linearity using analysis of variance (ANOVA), absence of multicollinearity with the variance inflation factor, independence of residuals with Durbin–Watson, normality of residuals, and homoscedasticity. Analyses were conducted using SPSS version 29, and significance was considered at a *p*-value < 0.05.

### 2.7. Ethical Considerations

This study received approval from the Research Bioethics Committee of the Universidad Cooperativa Colombia, under bioethical concept no. BIO528. All participants willingly agreed to take part in the study by providing their signature on the informed consent form.

## 3. Results

A total of 352 students were included in the study, with an average age of 22 ± 4.5 years. Most (70.7%) were female, and 52.6% were in the basic training cycle. About 31% rated the education received at the university regarding the relationship between health and air quality as either regular or poor. Concerning air quality in the city, only 17.3% expressed satisfaction, 81% perceived it as poor, and 83.2% believed it would not improve in the coming years (Table 1).

### 3.1. Knowledge

For the knowledge index, it was discovered that approximately 90% of medical students believed that exposure to air pollution increases the risk of complications in pregnancy and the fetus, inflammatory skin diseases, neurological diseases, cardiovascular diseases, and respiratory diseases. Regarding the sources of air pollution, it is noteworthy that 23% did not recognize that extensive livestock farming constitutes a source of air pollution. Concerning measures to reduce the harmful effects of pollution, 82.4% believed using face masks is a useful measure. In comparison, 29.8% did not find hydrating with water throughout the day beneficial, and 29.3% did not consider a healthy diet a useful measure (Figure 1). The median global score in this domain was 77.8 (IQR 71.1–85.6), and significant differences were only found according to age group (*p* = 0.034) and the perception of the education received on the topic (*p* = 0.037) (Table 2).

### 3.2. Attitudes

The attitudes index showed that about 95% of the students believed that air pollution is a problem affecting public health worldwide, as well as within the country and the city. Similarly, 96.6% thought that healthcare personnel should inform their patients about the effects of air pollution; however, 25.9% considered that air pollution is a multifactorial problem, so their actions would not have any impact. Consistently, 37.5% said they would not be willing to reduce their usual beef consumption, 36.9% said they would not be willing to reduce dairy consumption, and 32.1% said they would not be willing to use bicycles for transportation to protect air quality (Figure 2). The median attitude score was 82.1 (IQR 71.8–87.2), and women scored significantly higher (*p*-value 0.005) (Table 2).

### 3.3. Practices

Regarding practices, it was found that 47.2% had never or rarely investigated the health effects of air pollution, 19.3% had not recommended any measures to family or friends to protect themselves from air pollution, 22.2% had not taken measures to protect themselves, and 31.8% did not monitor air pollution indicators in the city. Concerning practices contributing to pollution, 27.6% consumed beef more than two times per week, 35.8% consumed dairy products more than three times per week, and only 33.2% used transportation means that do not rely on fossil fuels (Figure 3). The practices score was the lowest, with a median of 50 (IQR 42.9–57.1), and significant differences were observed based on the perception of education received (*p*-value 0.001), the perception of city air quality (*p*-value 0.046), and the satisfaction with city air quality (*p*-value 0.017) (Table 2).

### 3.4. Factors Associated with Knowledge, Attitudes, and Practices

Linear regression models were employed to identify the factors associated with knowledge, attitudes, and practices. The models revealed that knowledge was only associated with attitudes; attitudes were associated with gender, knowledge, and practices; while practices were associated with attitudes. No significant associations were found between knowledge and practices (Figure 4).

## 4. Discussion

Exposure to poor air quality is a significant public health issue, emerging as the foremost environmental threat to health, even surpassing the lack of clean water [35,36]. Numerous studies have established associations between air pollution and chronic respiratory diseases, cardiovascular conditions, neurological disorders, and complications during pregnancy and childbirth [6,12,13,15,17,18]. There is a consensus that if society were better informed about the link between poor air quality and adverse health outcomes and were aware of actions that could mitigate pollution, individual, societal, and policy changes could be prompted [37]. Healthcare professionals play a crucial role in supporting these actions, starting with being well-informed about air pollution sources and then incorporating this knowledge into their practice and daily patient interactions [29].

In our study, approximately 90% of medical students recognized the risks associated with exposure to air pollution and its impact on health (Figure 1). However, despite this awareness, four out of five medical students considered an ineffective measure, such as wearing face masks, helpful in reducing pollution’s harmful effects. This high level of knowledge aligns with other studies conducted in different populations, such as China, Ghana, Lebanon, the United States, Kuwait, Namibia, and Ireland, where a widespread acknowledgment of the hazards of air pollution was observed [29,32,33,34,38,39,40,41]. Regarding protective measures, studies have demonstrated that masks, especially N95 respirators, provide adequate protection approximately 74% of the time the respirator is worn, with a significant dependence on the type of respirator. However, loose-fitting masks and improvised devices hardly achieve the level of protection found with N95 respirators [42]. On the other hand, another study showed that, on average, only 1% of the time a surgical mask was used did it result in adequate levels of protection [43], reinforcing the notion that loose-fitting and improvised facial masks are not designed to create an airtight seal and therefore cannot prevent airborne particles from bypassing the filter and being inhaled into the respiratory tract during normal breathing, potentially causing harm [42]. Therefore, regarding the utilization of masks as a preventive measure, it is imperative to ensure that forthcoming medical professionals possess knowledge pertaining to diverse mask varieties and their efficacy and limitations to avoid disseminating inaccurate information to patients. Although our data do not permit definitive conclusions regarding the extent to which this knowledge is possessed by individuals, it is crucial to underscore that endorsing ineffective interventions, including mask usage, not only affects the educational trajectory of aspiring physicians but also risks perpetuating misguided notions among future patient cohorts if such beliefs persist.

On the other hand, the results of our study highlight trends in knowledge regarding the benefits of water hydration and a healthy diet in mitigating the health impacts of environmental pollution. Specifically, 29.8% of respondents did not see the value of staying hydrated throughout the day, while 29.3% did not consider a healthy diet beneficial. These findings underscore the crucial need for increased awareness and education about the importance of dietary habits in combating the adverse health effects of environmental pollutants [44]. Research consistently shows that adequate water intake plays a vital role in physiological processes essential for overall health, including detoxification, respiratory health, temperature regulation, and immune function, all contributing to mitigating the effects of pollution [45]. Similarly, a diet rich in fruits, vegetables, whole grains, and lean proteins provides essential nutrients like vitamins, minerals, and antioxidants, strengthening the immune system and combating oxidative stress induced by pollution [46]. Moreover, anti-inflammatory foods such as omega-3-rich fish can reduce inflammation exacerbated by environmental pollutants, contributing to cardiovascular and respiratory health [46,47]. Therefore, promoting healthy lifestyle practices, including regular water consumption and a balanced diet, is crucial in strategies to mitigate the health impacts of environmental pollution.

Regarding attitudes, 95% of the medical students in this study recognized that air pollution is a public health problem internationally and locally, as shown in Figure 2, consistent with findings from other studies [28,34,38,48] highlighting a widespread perception of the severity of air pollution. Despite this awareness, one in four students believed that their actions had no impact on air pollution and therefore said they would not be willing to modify behaviors to reduce it, such as decreasing the regular consumption of beef and dairy products or using alternative modes of transportation. Previous studies have identified air as a common resource [49]. The use of air generates individual benefits, but the damage to this shared resource is distributed among all potential users. Individuals conducting short-term cost–benefit analyses may conclude that individual efforts to protect air yield no immediate benefits. Consequently, several individuals act independently, driven by personal interest and comfort, thereby depleting the common-use resource. In this context, promoting collective action through efforts involving governments, international cooperation, academia, and civil society is necessary to reduce the emission of pollutants into the environment and raise awareness of individual contributions and responsibility in addressing this issue [50,51,52]. Gender emerges as an influential factor in relation to attitudes. For instance, various studies conducted with different populations indicate that women tend to have higher scores in attitudes, meaning they exhibit more favorable attitudes toward air pollution [39,41]. The findings in this study suggest that women had higher scores in attitudes. However, this was not related to practices, similar to what was reported in the study by Al-Khamees, where women scored significantly higher than men in attitudes but were also significantly more likely to engage in polluting practices [41].

The integration of healthy practices to mitigate the health effects of air pollution is crucial; however, in our study, the score for practices was the lowest and revealed behaviors contributing to air pollution, such as the use of aerosols for household cleaning, personal hygiene, and limited use of transportation not requiring fossil fuels. Similarly, it highlighted incorrect practices for protecting oneself from air pollution: one in five students did not adopt any measures to protect against pollution, 71.6% had “air-purifying” plants, and 25.3% used masks as a protective measure. These findings are consistent with other studies where suboptimal practices related to air pollution were observed in students [38,40]. Considering that environmental pollutants are emitted through various human activities practiced on a large scale, such as industrial machinery, power plants, and fossil fuel engines, their contribution is substantial, with automobiles being responsible for approximately 80% of current pollution [53]. Other activities that may be considered minor, such as farming techniques, gas stations, fossil fuel-based heaters, cleaning procedures, and aerosol substances for personal care, impact global pollution that must be prevented [54]. Therefore, the perception that some sources of pollution are minor does not imply that they can be deemed acceptable or that they should not be avoided, as these also contribute to air pollution [7]. This misguided perception of individual contributions can impact the future medical practice of medical students, as these practices, which should also be avoided, may be transmitted to future patients; hence the importance of education and its consistency with practices that can benefit the environment.

Although, theoretically, in KAP-type studies, there should be a positive correlation between levels of knowledge, attitudes, and practices, and some studies related to air pollution have demonstrated such associations [32,40], an interesting finding in our study was that, despite the high level of knowledge, this did not correlate with the level of practices. In other words, even though the participants had solid knowledge about air pollution, there needs to be more clarity regarding theoretical knowledge and actual behavior (Figure 4).

These findings could have different explanations. It is possible that acquired information does not directly translate into concrete behaviors because attitudes and practices may be rooted in cultural and social norms and these influences can be more determinant than the provided technical information. Additionally, there might be a perception that individual actions have a limited impact on improving air quality. Although participants may clearly understand recommended practices, they might need to be more confident in their ability to effect change or they might perceive the problem as too vast to address individually. The lack of correlation between knowledge and practices in KAP studies is partially explained by the “theory of planned behavior”, which highlights the role of behavioral intentions influenced by attitudes, social norms, and perceived control. However, several factors complicate this translation of intentions into actions. Firstly, while individuals may have a strong understanding of environmental issues like air pollution, their attitudes and beliefs, shaped by cultural and social factors, can significantly impact their behavior. Additionally, subjective norms, reflecting societal pressures, can influence behavior, especially if societal indifference towards sustainable practices is perceived. Moreover, perceived behavioral control, encompassing beliefs about one’s ability to act and external barriers, can hinder action despite good intentions [55]. The “environmental attitude–behavior gap” underscores this disconnect between intentions and actions despite theoretical frameworks like the theory of planned behavior. Addressing this gap requires considering practical barriers, societal norms, and individual beliefs, emphasizing the need for a holistic approach to addressing the knowledge–practice gap in air pollution mitigation [56]. Finally, decisions and practices may be strongly influenced by external factors, such as the availability of sustainable options, city infrastructure, and accessibility to resources facilitating the adoption of healthier behaviors. Adopting more sustainable practices could face practical and logistical barriers that need to be fully addressed with basic knowledge.

This study has the following limitations: (i) the design was cross-sectional; therefore, the associations do not have a causal nature; (ii) social desirability bias could have led the participants to respond according to a distorted image of what they know, believe, or do; (iii) although the study included a large sample of medical students, it was not representative of the entire city; (iv) the questionnaires lacked a deeper exploration of indoor air pollution sources, such as wood burning, a common practice in some regions of the country.

## 5. Conclusions

Regarding the knowledge scale, the results of this study highlighted that medical students answered the majority of questions regarding the sources and effects of air pollution on health correctly. However, they revealed gaps in knowledge or misconceptions about practices to protect themselves from air pollution. Additionally, the practices used to protect against pollution were found to be ineffective. The lack of a direct correlation between knowledge and practices identified in this study underscores the need to strengthen environmental health education through a more comprehensive approach that encompasses normative and cultural actions. This should focus on enhancing the practical application of acquired knowledge. These young healthcare professionals will guide and advise their patients on preventive measures and healthy lifestyles. Therefore, they must be well-informed and develop proactive attitudes and effective practices to address the challenges of air pollution. Implementing changes in educational curricula, explicitly focusing on environmental health, climate change, and strategies to tackle air pollution, is crucial to equip these future doctors with the necessary tools. Air pollution, driven by emissions from various sources, releases harmful pollutants into the atmosphere. These pollutants not only contribute to the greenhouse effect, leading to climate change, but also directly impact human health by causing respiratory diseases, cardiovascular problems, and other health issues. Additionally, climate change amplifies the effects of air pollution, creating a feedback loop that further deteriorates air quality and exacerbates health risks for populations worldwide. Understanding and addressing this complex interplay is crucial for safeguarding public health and mitigating the adverse impacts of environmental degradation. By recognizing the interconnectedness between human and environmental health through the concept of “One Health”, the importance of integrating environmental awareness into everyday medical practice is underscored. Medical students and healthcare professionals should be agents of change, advocating for public health policies that promote cleaner and healthier environments. As we move towards the future, global health will depend significantly on the ability of healthcare professionals to address environmental complexities and promote practices that contribute to a healthier and more sustainable world [57,58].

## Figures and Tables

**Figure 1 ijerph-21-00789-f001:**
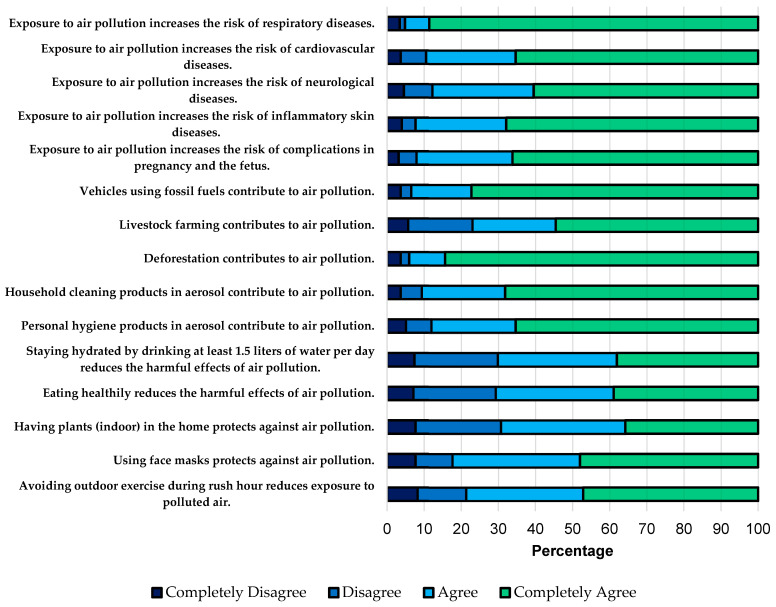
Relative frequencies of responses to knowledge items on air pollution.

**Figure 2 ijerph-21-00789-f002:**
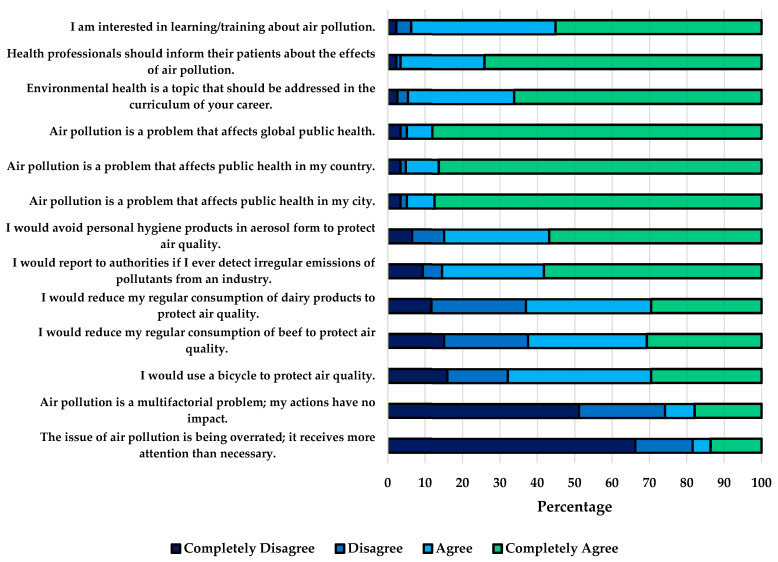
Relative frequencies of responses to items on attitudes towards air pollution.

**Figure 3 ijerph-21-00789-f003:**
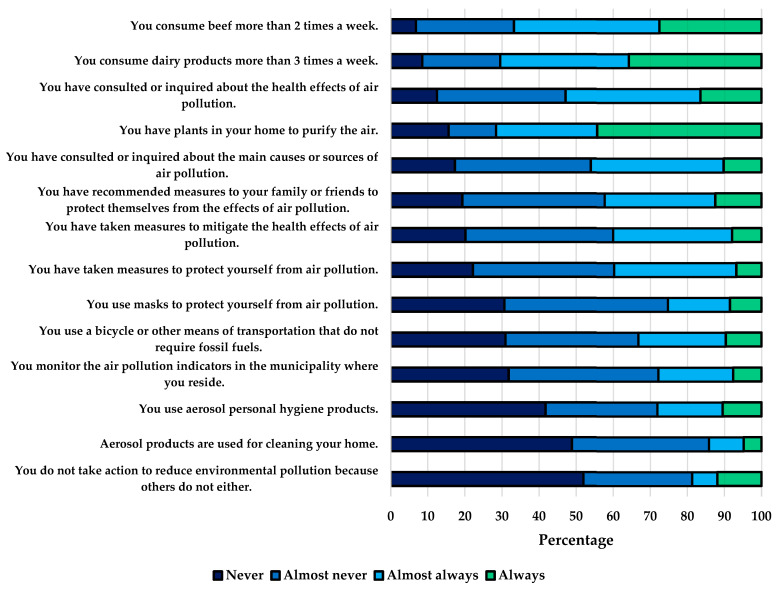
Relative frequencies of responses to items on practices related to air pollution.

**Figure 4 ijerph-21-00789-f004:**
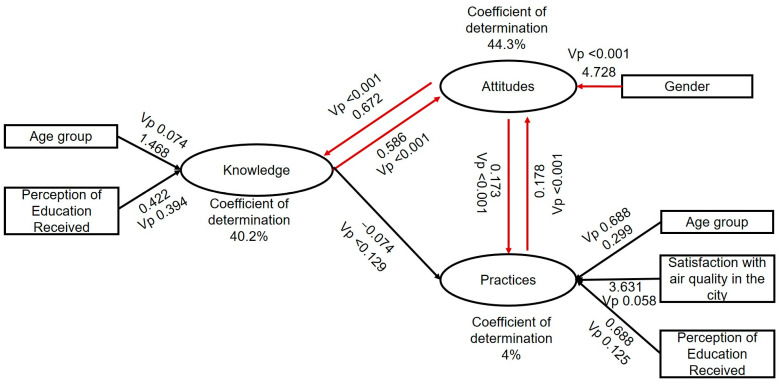
The figure displays the regression coefficients of the factors associated with knowledge, attitudes, and practices. For all variables, the *p*-values were less than 0.05. The red lines represent significant associations. The black lines represent confounding variables, variables that were associated in the bivariate analysis but whose association disappeared in the multivariate analysis.

**Table 1 ijerph-21-00789-t001:** Description of demographic and educational characteristics of the study group.

		*n*	%
Age Group	18 to 20 years	127	36.1
	21 to 23 years	134	38.1
	Older than 23 years	91	25.9
Gender	Female	249	70.7
	Male	103	29.3
Monthly Family Income	Less than 1 minimum wage	15	4.3
	Between 1 and 2 minimum wages	100	28.4
	Between 3 and 4 minimum wages	121	34.4
	More than 4 minimum wages	116	33.0
Training Cycle	Basic (I to V)	185	52.6
	Clinical (VI to X)	129	36.6
	Professional or medical internship (XI-XII)	38	10.8
Perception of Education Received on Health and Air Quality Relationship	Not received	109	31.0
	Excellent	28	8.0
	Good	106	30.1
	Regular	90	25.6
	Poor	19	5.4
Are You Satisfied with the City’s Air Quality?	No	291	82.7
	Yes	61	17.3
Perception of Air Quality in the City (Previous Year)	Excellent	7	2.0
	Good	61	17.3
	Regular	205	58.2
	Poor	79	22.4
Perception of Current Air Quality in the City	Excellent	7	2.0
	Good	60	17.0
	Regular	198	56.3
	Poor	87	24.7
Considers that Air Quality in the City Will Improve in the Coming Years	No	293	83.2
	Yes	59	16.8

**Table 2 ijerph-21-00789-t002:** Comparison of the demographic and educational characteristics of the study group with knowledge, attitudes, and practices.

		Knowledge	Attitudes	Practices
		Me (IQR)	Me (IQR)	Me (IQR)
Age Group ^b^	18 to 20 years	77.8 (71.1–84.4)	82.1 (74.4–87.2)	50.0 (42.9–57.1)
	21 to 23 years	75.6 (68.9–84.4)	79.5 (71.8–87.2)	47.6 (40.5–54.8)
	Over 23 years	82.2 (73.3–86.7)	84.6 (74.4–87.2)	50.0 (42.9–57.1)
	*p*-value	0.034 *	0.184	0.280
Gender ^a^	Female	77.8 (68.9–86.7)	82.1 (74.4–87.2)	50.0 (42.9–57.1)
	Male	77.8 (71.1–84.4)	76.9 (69.2–84.6)	47.6 (40.5–54.8)
	*p*-value	0.966	0.005 **	0.158
Monthly Family Income ^b^	Less than 1 minimum wage	82.2 (71.1–86.7)	79.5 (66.7–84.6)	50.0 (42.9–59.5)
	Between 1 and 2 minimum wages	75.6 (68.9–83.3)	79.5 (71.8–84.6)	50.0 (41.7–57.1)
	Between 3 and 4 minimum wages	80.0 (68.9–86.7)	82.1 (74.4–87.2)	47.6 (42.9–57.1)
	More than 4 minimum wages	77.8 (71.1–84.4)	82.1 (73.1–89.7)	48.8 (41.7–54.8)
	*p*-value	0.481	0.191	0.692
Training Cycle ^b^	Basic (I to V)	77.8 (71.1–84.4)	79.5 (71.8–87.2)	47.6 (42.9–57.1)
	Clinical (VI to X)	77.8 (71.1–84.4)	82.1 (74.4–87.2)	50.0 (40.5–57.1)
	Professional or medical internship (XI-XII)	81.1 (68.9–86.7)	82.1 (74.4–89.7)	52.4 (45.2–57.1)
	*p*-value	0.352	0.416	0.504
Perception of Education Received on Health and Air Quality Relationship ^b^	Not received	86.7 (76.7–86.7)	84.6 (75.6–88.5)	52.4 (46.4–58.3)
	Excellent	80.0 (71.1–84.4)	83.3 (74.4–87.2)	52.4 (45.2–59.5)
	Good	77.8 (71.1–86.7)	80.8 (71.8–87.2)	50.0 (42.9–57.1)
	Regular	77.8 (62.2–84.4)	79.5 (64.1–84.6)	42.9 (35.7–52.4)
	Poor	77.8 (68.9–84.4)	79.5 (69.2–89.7)	47.6 (40.5–52.4)
	*p*-value	0.037 *	0.295	0.001 **
Are You Satisfied with the City’s Air Quality? ^a^	No	77.8 (71.1–84.4)	82.1 (71.8–87.2)	50.0 (42.9–57.1)
	Yes	80.0 (68.9–86.7)	79.5 (71.8–84.6)	45.2 (38.1–54.8)
	*p*-value	0.610	0.068	0.017 *
Perception of Air Quality in the City (Previous Year) ^b^	Excellent	84.4 (71.1–86.7)	84.6 (71.8–100.0)	52.4 (42.9–69.0)
	Good	77.8 (68.9–84.4)	79.5 (71.8–84.6)	47.6 (38.1–54.8)
	Regular	77.8 (71.1–84.4)	82.1 (74.4–87.2)	50.0 (42.9–57.1)
	Poor	77.8 (68.9–86.7)	79.5 (69.2–89.7)	50.0 (40.5–57.1)
	*p*-value	0.643	0.265	0.214
Perception of Current Air Quality in the City ^b^	Excellent	84.4 (84.4–86.7)	76.9 (64.1–92.3)	52.4 (42.9–69.0)
	Good	82.2 (70.0–86.7)	79.5 (71.8–84.6)	45.2 (38.1–52.4)
	Regular	77.8 (71.1–84.4)	82.1 (74.4–87.2)	50.0 (42.9–57.1)
	Poor	77.8 (68.9–86.7)	82.1 (69.2–89.7)	50.0 (42.9–59.5)
	*p*-value	0.123	0.110	0.046 *
Considers that Air Quality in the City Will Improve in the Coming Years ^a^	No	77.8 (71.1–86.7)	82.1 (71.8–87.2)	50.0 (42.9–57.1)
	Yes	77.8 (68.9–84.4)	82.1 (74.4–87.2)	50.0 (42.9–59.5)
	*p*-value	0.764	0.684	0.177

Me: Median. IQR: Interquartile range. ^a^ Mann–Whitney U test. ^b^ Kruskal–Wallis H test. * *p*-value < 0.05, ** *p*-value < 0.01.

## Data Availability

The datasets analyzed during the current study are available from the corresponding author upon reasonable request or via the following link: https://papyrus-datos.co/dataverse/conocimientos (accessed on 23 April 2024).

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
