# Peer review of "Knowledge, Attitudes, and Practices Regarding Air Pollution among Medical Students"

_ijerph, 2024, doi:10.3390/ijerph21060789_

Round 1
Reviewer 1 Report
Comments and Suggestions for Authors
Generally this is an interesting study. I agree that medical students, and health professionals in general, have a role to play in educating their patients, and their communities, about the health effects of air pollution; and measure they can take to reduce emissions, and to protect themselves.
Medellin certainly has a significant air pollution problem. I think the intro should include some info about not just the city; but the country; as students will tend to work both urban and rurally. (line 57: In recent years, the city of Medellín, Colombia, has witnessed recurring episodes of air pollution). It would give better context.
3.3; line 200; Never inquired; please be clear; inquired to whom; to patients? Do you mean when taking a clinical history, in clinical encounters?
my main concern with the paper is the lack of clarity about sources of air pollution.
Line 44; it is also imperative to consider indoor air pollution originating from conventional and electronic cigarettes, organic solvents in household and personal care products, heating, wood burning, and other domestic consumer goods
Line 57-70 is a very good paragraph with a clear description of the main sources in the city.
but later the focus moves off these major emission sources; and mentions livestock farming. Is this a major issue in Medellin?
Even rurally; the issue of most concern would likely be household air pollution ie burning of solid fuel, and or kerosene, for cooking and heating and lighting.
Livestock farming is definitely an important issue in climate chnage, with methane emissions; but less so as air pollution; although methane might contribute to ozone formation. If so, in Medellin, please show a reference for this
You need to show evidence that these questions are relevant; . In comparison, 29.8% do not find hydrating with water throughout the day beneficial,(is that relevant to air pollution; or heat?) and 29.3% do not consider a healthy diet as a useful measure. That might be relevant in that some vitamins /antioxidants show some beneficial effect; but you need to cite the evidence to make that question relevant.
Also, re sources, Line 44; "it is also imperative to consider indoor air pollution originating from conventional and electronic cigarettes, organic solvents in household and personal care products, heating, wood burning, and other domestic consumer goods"
I agree that cigarettes is important, So is heating, through burning wood or other solid fuels (coal, charcoal, dung). Please show data about how many households burn wood in Medellin; or can reference Columbia as a whole; where it is a significant problem.
I do not see any good refs for including "organic solvents in household and personal care products, and other domestic consumer goods" and I think if you look at the literature, these indoor sources will be relatively insignificant. There is recent literature about NO2 from gas stoves indoors.
These questions re these less important sources are in the data. Please justify them or remove them.
It would also be useful to point out the difference between: sources such as traffic, industrial emissions: need a societal, community response; vs indoor sources like smoking that need an individual response, or a public health campaign.
line 299; exciting finding; suggest another word instead of exciting
Line 335: "Medical students should be agents of change, advocating for public health policies that promote cleaner and healthier environments".
there is a good opportunity here to mention links to climate change. There are strong links between air pollution and climate change and health eg active transportation to reduce air pollution reduces GHGs, and also improves health. Id suggest while discussing "one health" etc, you also mention these links to GHG emissions and climate chnange, and the potential role of health professionals in this
Author Response
Dear Reviewer: Please see the attachment

Reviewer 2 Report
Comments and Suggestions for Authors
Please see attached file.

Overall, language seems fine, but I am not a native speaker. See in the attached file the small typos that I noticed.
Author Response

(The authors gave the same response as above.)
